# Long-Term Clinical Outcomes and Carotid Ultrasound Follow-Up of Transcarotid TAVI. Prospective Single-Center Registry

**DOI:** 10.3390/jcm10071499

**Published:** 2021-04-04

**Authors:** Damian Hudziak, Adrianna Hajder, Radoslaw Gocol, Marcin Malinowski, Maciej Kazmierski, Lukasz Morkisz, Joanna Ciosek, Wojciech Wanha, Grzegorz Jarosinski, Radoslaw Parma, Tomasz Darocha, Wojciech Wojakowski

**Affiliations:** 1Department of Cardiac Surgery, Medical University of Silesia, 40-007 Katowice, Poland; gocot@poczta.onet.pl (R.G.); marmal@interia.pl (M.M.); lukasz.morkisz@gmail.com (L.M.); 2Division of Cardiology and Structural Heart Diseases, Medical University of Silesia, 40-007 Katowice, Poland; kazmierski.maciej@gmail.com (M.K.); joanna.ciosek@gmail.com (J.C.); wojciech.wanha@gmail.com (W.W.); radoslaw.parma@gmail.com (R.P.); wojtek.wojakowski@gmail.com (W.W.); 3Department of Electrocardiology and Heart Failure, Medical University of Silesia, 40-007 Katowice, Poland; gregjaro@gmail.com; 4Department of Anaesthesiology and Intensive Care, Medical University of Silesia, 40-007 Katowice, Poland; tomekdarocha@wp.pl

**Keywords:** transcatheter aortic valve implantation, aortic valve stenosis, transcarotid access, oximetry

## Abstract

This study aimed to prospectively evaluate the safety and long-term clinical outcomes of cerebral-oximetry-guided transcarotid transcatheter aortic valve implantation (TC-TAVI) with systematic follow-up with carotid ultrasound. Thirty-three TCTAVI procedures were performed in our center from 2017 to 2019. Our analysis includes in-hospital outcomes and long-term follow-up data on mortality, echocardiographic parameters, carotid Doppler ultrasound, and VARC-2 defined clinical events. Intraoperatively, one patient died, and one had a transient ischemic attack (TIA). The following events occurred in-hospital postoperatively: myocardial infarction (3.0%), cardiac tamponade (3.0%), new-onset atrial fibrillation (6.3%), need for temporary pacing (27.3%) and need for pacemaker implantation (15%). The mean follow-up was 19.5 ± 9.52 months. In the long-term follow-up, the two-year survival rate was 83% ± 14. The echocardiographic parameters did not differ significantly from the postprocedural values, and the ultrasound did not show any cases of significant vessel narrowing. The mean peak systolic velocity (PSV) was 71.6 cm/s in the left common carotid artery and 70.6 cm/s in the right common carotid artery. In conclusion, cerebral oximetry-guided TC access is safe, has a favorable long-term outcome, and does not increase the risk of plaque formation in the carotid artery. In a carefully selected group of patients, it might be considered as a first-choice alternative to TF access.

## 1. Introduction

One of the main goals of TAVI operators is to reduce the risk of vascular complications (VC), which frequently occurred during the early days of the TAVI experience. The complications were mostly related to suboptimal aortic and iliofemoral anatomy in high-risk patients. With new valve designs, the VC rate in patients treated with transfemoral (TF) access declined significantly to less than 10% in current registries. However, in high-risk patients, the extent of calcifications, a small diameter of the common femoral and iliac arteries, and diffuse atherosclerosis preclude a safe TF procedure. The use of alternative access routes has allowed TAVI to be carried out in patients in whom TF could not be performed. Such alternative access techniques include the transapical (TA), trans-axillary, direct aortic, trans-caval, and trans-carotid (TC) approach. The TA approach was the first to be used, and the clinical evidence related to the alternative TAVI access is related mostly to this approach. Several observational studies showed that when feasible, the TF approach is the safest option with significantly fewer complications than the surgical TA one. TC-TAVI, when done by an experienced operator, is a safe alternative to TF and allows faster patient ambulation than the TA route. TC-TAVI can be done using different transcatheter valve systems [1], conscious sedation in selected patients, and when used with monitoring of cerebral oximetry, has a low risk of neurological complications [2]. Several registries showed favorable outcomes of TC-TAVI; however, prospective data on long-term results are limited. The current paper provides long term outcomes with routine follow-up with carotid duplex ultrasound (DUS) of all patients treated with TC-TAVI in a large academic institution where TC is the preferred alternative access when TF is not feasible. This is a long-term extension of our previously published data with systematic follow-up with carotid ultrasound [1,2,3].

## 2. Materials and Methods

The methods and design of the registry were previously published [2,3]. In this prospective study, we enrolled consecutive patients with severe symptomatic aortic stenosis who were treated in our hospital with TC-TAVI between 2017 and 2019. All patients provided written informed consent to undergo the TAVI procedure according to eligibility evaluation. No institutional review board or ethics committee approval was required for this study. The results were presented in line with the Valve Academic Research Consortium-2 (VARC-2) consensus [4].

The Heart Team decided the access route used for valve implantation. The decision was based on the findings from multislice computed tomography (MSCT) of the aorta and peripheral arteries, transthoracic echocardiography, and coronary angiography. The 3Mensio program (Pie Medical Imaging, Bilthoven, The Netherlands) was used for the analysis of scans. Patients with peripheral artery disease (PAD), extensive calcifications, critical iliac artery stenosis, or extreme tortuosity, as well as a small iliac artery diameter (<⁠6 mm), were deemed ineligible for TFTAVI. In addition, significant aortic disease at the level of the thoracic and abdominal aorta, such as aneurysm, presence of an aortic stent graft, large thrombus, chronic aortic dissection, or excessive aortic tortuosity, favored the choice of TC access. Based on the analysis of MSCT scans (Figure 1A), patients had to have a carotid artery diameter of at least 5.5 mm, no significant ipsi- or contralateral (>50%) common carotid artery (CCA) stenosis, and no significant calcifications in order to be eligible for TC-TAVI. 

The type and size of the valve prosthesis was selected on the basis of the MSCT and the clinical factors, such as the risk of conduction disturbances. The following MSCT parameters were taken into consideration: diameters of left and right CCA, occurrence of bicuspid aortic valves (BAV), height of left main coronary artery (LM) and right coronary artery (RCA), aortic valve annulus area (AVAA) and the degree and distribution of calcifications in the aortic valve and in the aortic annulus. A detailed analysis of the MSCT allows one to choose an artery with a larger diameter, less calcification and tortuosity, and with a more favorable spatial relationship between the virtual CCA center line and the plane of the aortic annulus. In the case of borderline dimensions of the CCA, the sheathless technique for the self-expandable valve was preferred. On the other hand, a high risk of conduction disturbances and a high probability of future coronary interventions were considered an indication for a balloon expandable valve.

All procedures were performed by the Heart Team, in a hybrid room and under general anesthesia. All patients were given antibiotics (cephazolin 1.5 g) as prophylaxis of infectious endocarditis. The CCA was carefully dissected, especially protecting the vagus nerve, and manually examined after dissection for the presence of calcifications. Two 5-0 monofilament continuous purse-string sutures were used similarly to the technique used to cannulate the ascending aorta during the classical surgical operation. After the procedure, the carotid artery was closed using sutures without clamping. We prefer not to clamp the carotid artery to reduce its trauma (Figure 1B). After placing a vascular access catheter (6F) in the carotid or femoral artery, heparin was administered (100 U/kg; activated clotting time > 250 s). Patients were monitored during the whole procedure, and the following parameters were recorded: arterial blood pressure and central venous pressure, evaluation of arterial blood saturation, and electrocardiogram (ECG). In addition, cerebral oximetry (INVOS 5100C, Medtronic, Dublin, Ireland) was monitored during TC-TAVI. In the majority of procedures, transthoracic echocardiography was performed. In selected cases, transoesophageal echocardiography (TEE) was also used. The electrode for pacing was placed using a 6F sheath through the internal jugular vein or the femoral vein. Follow-up arteriography was carried out after removing the whole system from the carotid artery (Figure 1C). Carotid duplex ultrasound imaging (DUS) was performed according to the published consensus recommendations. The study included greyscale 2D, spectral, and color Doppler imaging. The peak systolic velocity (PSV) and the end-diastolic velocity (EDV) were assessed as well as semiquantitative plaque assessment [5,6] (Figure 1D).

### Statistical Methods

Data are presented as mean (SD) when normally distributed or as median with 25th and 75th percentiles when normality assumptions (Shapiro–Wilk test) were not met. Categorical data are expressed as a percentage. Paired t-test or Wilcoxon Signed Rank test was used to compare data before and after the procedure. Two-way Repeated Measure Analysis of Variance (RM ANOVA) with Holm–Sidak post hoc tests was carried out to test for the effect of time and side on cerebral oximetry. One way RM ANOVA with Holm-Sidak post hoc test was used to test for differences between data before and various time points after the procedure. The probability of survival (with 95% CI) is presented on a Kaplan–Meier curve. All statistical analysis was performed with SigmaPlot ver. 12.5 (Systat Software, San Jose, CA, USA) and GraphPad Prism ver. 9.0 (GraphPad Software, La Jolla, CA, USA) was used to construct K-M survival curve. *p*-value of less than 0.05 was considered significant.

## 3. Results

### 3.1. Patients’ Characteristics

Thirty-three TC-TAVI procedures performed in the Upper-Silesian Medical Center of the Medical University of Silesia in Katowice, Poland, from September 2017 to November 2019, for whom follow-up of at least 12 months is available were included. In these years we performed 232 TF-TAVI, 33 TC-TAVI and 9 TA-TAVI procedures. The TC-TAVI are 12.0% of all TAVI procedures in our center. The baseline demographic and echocardiographic characteristics are included in Table 1 and Table 2. The median age was 77 (72–85), and 51.5% of patients were male. All patients were symptomatic (III-IV New York Heart Association (NYHA) functional class) and had echocardiographically confirmed severe aortic stenosis (median aortic valve area (AVA) 0.7 (0.6–0.8) cm^2^, median aortic valve mean gradient (Pg mean) 42 (35.5–54.5) mmHg). The median calculated risk of mortality, according to EuroSCORE II, was 6.1% (4.8–10.7). PAD and abnormalities of the descending aorta (thrombus, aortic aneurysm, history of intravascular or surgical intervention) occurred in 36.4% and 60% of patients, respectively. The decision to use TC access rather than transfemoral was made during the Heart Team discussion based on the presence of aortic disease or PAD and suboptimal anatomy and size of the iliofemoral vessels in the MSCT. In the TC-TAVI cohort, the median diameter of the femoral artery (FA) measured by MSCT was 5.7 (4.8–6.8) mm for the left and 5.4 (4.9–7.1) mm for the right side. 

### 3.2. Procedural and In-Hospital Outcomes

#### 3.2.1. Procedural Data

Self-expanding (Evolut R (Medtronic, Minneapolis, MN, USA), Portico (Abbott Vascular, Santa Clara, CA, USA)), and balloon-expandable valve prostheses (Edwards-Sapien 3 Ultra (Edwards Lifesciences Corp. Irvine, CA, USA)) were implanted (Table 3). Perioperative and early postoperative results are shown in Table 4. All of the TC-TAVI procedures were made under general anesthesia. The time of the procedure and mechanical ventilation was 65 (60–80) min, and 4 (1.5–6.5) hours, respectively. Most of the implantations (94%) were made via the left common carotid artery (LCCA). The median MSCT-measured diameter of LCCA was 6.2 (5.8–7.2) mm. All balloon-expandable valves were implanted through a dedicated vascular sheath, while in the majority of the self-expandable valves group, the sheath-less approach was used. Sheaths were used in eight patients only. All valve sizes were used, as shown in Table 3. 

#### 3.2.2. Periprocedural Outcomes

One patient died intraoperatively prior to prosthesis implantation due to aortic annulus rupture during balloon predilatation (procedural mortality 3%). In all of the remaining patients, the device was implanted as intended. Neurological complications occurred in 3.0% of patients treated by TC-TAVI (no strokes, one transient ischemic attack (TIA)). Results of bilateral cerebral oximetry during the procedure are shown in Figure 2. There were no differences in oximetry values at the baseline after CCA cannulation and postimplantation between the ipsi and contralateral side. A transient reduced oximetry value on the implantation side (42.1% ± 10.3 vs. 47.8% ± 10, *p* < 0.001) was observed only during rapid pacing. There was no significant difference between pre-and postprocedural cerebral oximetry values on the implantation side (69.1% ± 9.4 vs. 66.4% ± 8, *p* = 0.058). We observed one case (3.0%) of myocardial infarction and one case of heart tamponade (3.0%). No other events defined by VARC-2 consensus (prosthesis dislocation and dysfunction, life-threatening and major bleeding, major vascular complication, and acute renal injury) occurred. The percentage of new-onset atrial fibrillation was 6.3%. Nine patients (27.3%) required temporary pacing postoperatively, but the conduction disturbances resolved in 4 patients, and a permanent pacemaker was eventually implanted in 5 patients (15%). The median time of intensive unit care and total hospitalization was respectively 2 (2–3) and 6 (6–7) days. After the procedure, we noticed a reduction of the NYHA functional class and improvement of the echo parameters (AVA, mean PG, maximal PG, and transaortic peak instantaneous velocity (Vmax)) (Table 2). The preoperative and postoperative kidney function markers (level of creatinine and glomerular filtration rate (GFR)) were comparable (1.08 (0.87–1.28) vs. 1.05 (0.86–1.35), *p* = 0.432; 60.0 (44.5–70.0) vs. 57.0 (44.2–77.0), *p* = 0.603 respectively) (Table 1 and Table 4). In blood test results, we observed lower postoperative levels of hemoglobin and hematocrit (11.3 (9.9–11.9) vs. 12.9 (11.4–14.3), *p* < 0.001 and 34.0 (30.1–35.6) vs. 38.9 (34.4–42.2), *p* < 0.001 respectively). Wound complications (hematoma and infection) were not observed in the study group. In addition to one intraprocedural death, one patient died after the hospital discharge (30-day mortality 6.1%, *n* = 2). The cause of death in this patient was unknown.

### 3.3. Follow-Up Outcomes

Our analysis includes follow-up data on mortality, echo parameters, Doppler ultrasound of the carotid arteries, and VARC-2 defined clinical events.

#### 3.3.1. Clinical Outcomes

The mean follow-up was 19.5 ± 9.52 months. Seven (21%) deaths were noted during follow-up (2 (6.1%) stroke-related, 2 (6.1%) patients died from complications of COVID-19 infection, and causes of death in 3 patients are unknown). Three (12.1%) cerebrovascular events (two strokes and two TIA) occurred during the follow-up. All strokes occurred in patients with permanent atrial fibrillation. The estimated one-year Kaplan–Meier survival rate was 88% ± 11, and the two-year survival rate was 83% ± 14 (Figure 3). The median survival time was 34.8 months. The NYHA functional class was comparable to the NYHA class after the procedure (1(1–2) vs. 2(1–2), *p* = 0.138). The percentage distribution of NYHA class in long-term follow-up is presented in Table 4.

#### 3.3.2. Echocardiographic Follow-Up

Mean echo follow-up was 330 ± 140 days after the procedure. The control echo parameters did not differ from the postprocedural values, and there were no cases of abnormal transvalvular gradients or worsening of paravalvular leak (PVL) (Table 1).

#### 3.3.3. Carotid Doppler Ultrasound

In the ultrasound examination of the carotid arteries, there were no cases of significant vessel narrowing. The PSV values were below 125 cm/s, and the EDV was also normal in all patients (Table 4) (Figure 1D). 

## 4. Discussion

The current registry shows that TC-TAVI can be used as the first-choice alternative to TF with favorable clinical, echocardiographic and vascular outcomes. As a safety measure, all patients were monitored throughout the procedure with cerebral oximetry and in the long-term with carotid DUS. The study group included patients with severe symptomatic AS (EuroSCORE II 6.1%) who, by Heart Team discussion, were ineligible for TF access. The main reasons for alternative access were diffuse iliofemoral atherosclerosis, small vessel diameter, and less frequently, significant aortic disease. TC access was used after the evaluation of carotid arteries with MSCT. The periprocedural outcomes showed safety in this group of patients with multiple comorbidities and diffuse atherosclerosis. We observed one procedural death that was not related to the access route (annulus rupture) and one in-hospital TIA. Our protocol mandates the use of cerebral oximetry during the procedure. It showed a short period of reduced oxygenation on the access side during the rapid pacing only, but not cannulation itself. In addition, we prefer to use the self-expandable valves without the in-line sheath engaged in reducing the profile of the delivery system and maintaining the flow in CCA. Both balloon-expandable and self-expandable valves can be used through TC access [1]. The device success rate was 100%, and there were no increased rates of ≥PVL, permanent pacemaker implantation compared to patients undergoing TF-TAVI. Our analysis includes long-term follow-up data on mortality, echo parameters, Doppler ultrasound of the carotid arteries, and VARC-2 defined clinical events. The 1 and 2-year death-free survival was 88 and 83%, respectively. Importantly two of the patients died because of the complications of COVID19. In addition, we performed long-term follow-up with a carotid ultrasound to exclude the late complications of TC-TAVI access in all surviving patients. It showed that no new-onset stenotic lesions in CCA were detectable, which is reassuring. No similar data were published. It seems that the late effects of TAVI on the vasculature are related to the functional properties of the arteries, such as persistently increased arterial stiffness rather than a change of the plaque burden [7]. There were 2 stroke-related deaths and one TIA during follow-up, and both strokes occurred in patients with permanent atrial fibrillation and unknown anticoagulant status. The median survival time was 34.8 months, and the improvement in NYHA functional class was maintained during follow-up. The follow-up echo parameters did not differ from the postprocedural values, and there were no cases of abnormal transvalvular gradients or worsening of PVL. Recent propensity-matched analysis of 220 TF and 32 TC-TAVI patients showed similar 30-day mortality of 5% and 2.5% stroke rate. The outcomes were comparable after a median of 21 months in terms of death, myocardial infarction, and stroke [8]. Similarly, contemporary systemic review and meta-analysis including studies reporting 30-day outcomes, showed that TC-TAVI is associated with 5.3% mortality, 3.4% rate of stroke/TIA, 15.3% rate of permanent pacemaker and 2.4% of major VC. Importantly, time-sensitive analysis suggests the temporal trend of improved outcomes in terms of safety in centers performing TC-TAVI [6]. Data from the prospective FRANCE TAVI registry, including 435 patients treated with TC-TAVI, showed that this approach was used in 4% of all TAVI cases. Patients in need of an alternative to TF access were of higher surgical risk, but their 2-year mortality was similar to those treated through femoral access. Interestingly, this registry showed an increased risk of stroke (OR 2.42) and bleeding (OR 2.0) compared to TF, but not of vascular complications [9,10]. 

For patients in whom TF-TAVI is not feasible, there is a need to develop safe and reproducible alternative TAVI accesses. Until 2017, the first-choice alternative in our center was the transapical (TA-TAVI) approach. However, due to the invasiveness and increased risk of complications, we changed the strategy and adopted a less invasive alternative to this access. 

Our previously published data suggest that transcarotid access is comparable in outcomes to TF in properly selected patients and can be safely used as the first-choice alternative access in patients in whom the TF approach is not feasible [3]. In our experience, it is also safe to use local anesthesia as suggested in previously published studies [11]. Recently we finalized the multi-center registry comparing two alternative accesses: TC-TAVI vs. TA-TAVI.

## 5. Conclusions

In properly selected patients who cannot undergo TF-TAVI, an alternative approach of cerebral oximetry guided TC access is safe and has favorable long-term outcomes. The study confirmed that TC access does not increase the risk of plaque formation in the carotid artery in the long term.

## 6. Limitations

The study has limitations related to the sample size, single-center design, and use of different types of valves. However, in most centers, the TF approach is used in the majority of patients because of improvements in transcatheter valve technologies and the use of the alternative access reflects the real-life scenario in the valve center.

## Figures and Tables

**Figure 1 jcm-10-01499-f001:**
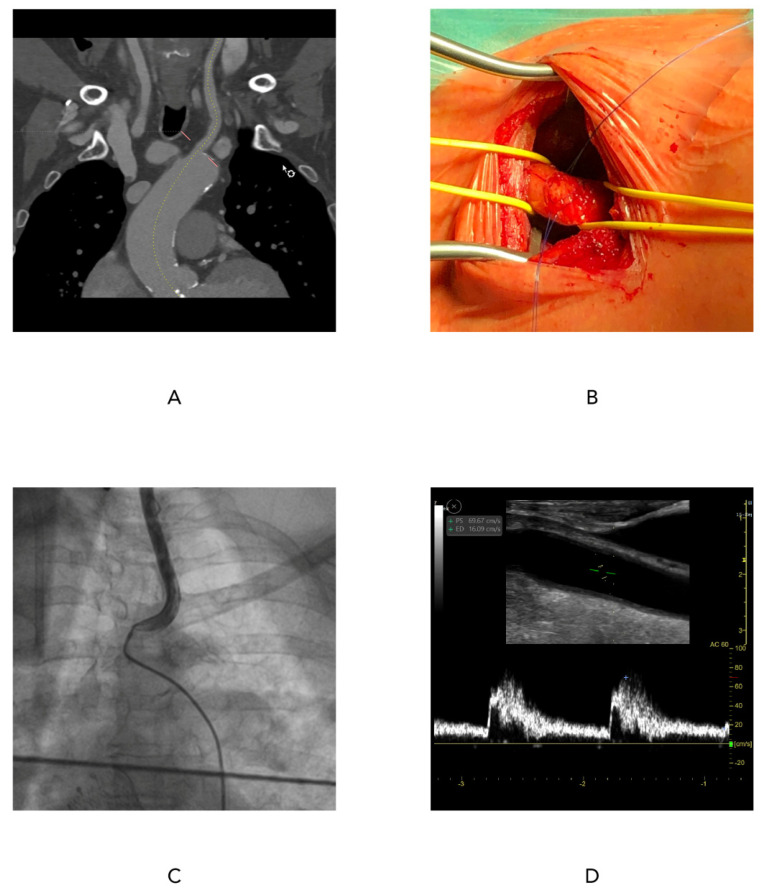
Procedural features. (**A**) Planning with MSCT; (**B**) Surgical preparation of the access site; (**C**) Post-implantation angiography; (**D**) Carotid Doppler ultrasound.

**Figure 2 jcm-10-01499-f002:**
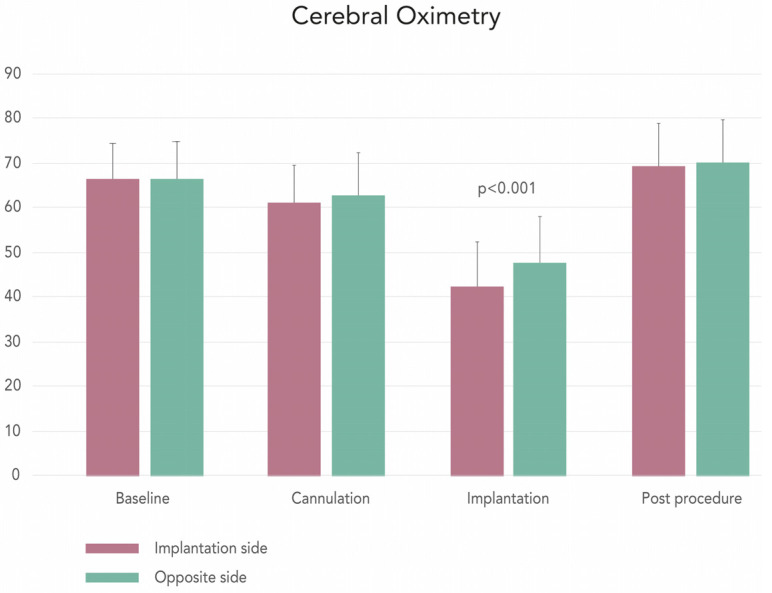
Cerebral oximetry.

**Figure 3 jcm-10-01499-f003:**
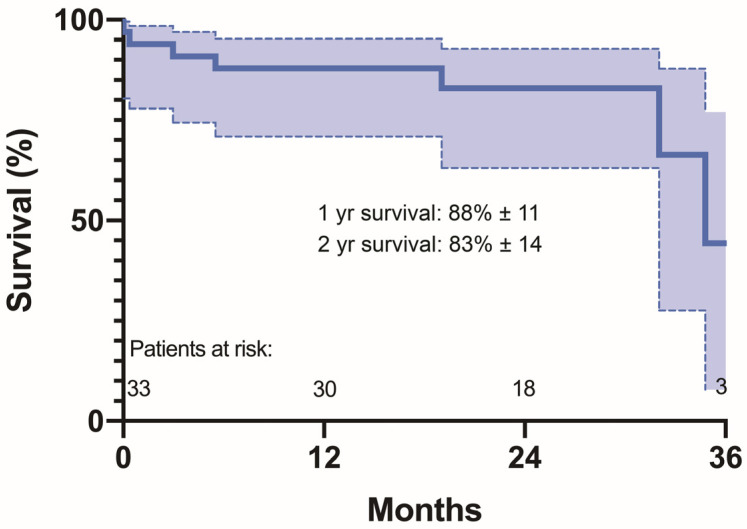
Kaplan–Meier survival curves.

**Table 1 jcm-10-01499-t001:** Baseline characteristics.

Variables	TC-TAVI (*n* = 33)
Age, years	77 (72–85)
Males	17 (51.5)
BMI, kg/m^2^	28 (24.0–31.5)
NYHA class	3 (3–3)
NYHA I	0 (0)
NYHA II	0 (0)
NYHA III	27 (81.8)
NYHA IV	6 (18.2)
EuroSCORE II, %	6.12 (4.8–10.7)
Cardiac comorbidities	
Prior MI	11 (33.3)
Prior cardiac surgery	10 (30.3)
Prior PCI	14 (42.4)
Atrial fibrillation	9 (27.3)
Pacemaker	1 (3.0)
Hypertension	33 (100)
Other comorbidities	
Dyslipidemia	24 (72.7)
Diabetes mellitus	19 (57.6)
COPD	8 (24.2)
PAD	12 (36.4)
DAD	20 (60)
Prior stroke	1 (3.0)
Prior TIA	0 (0)
Creatinine level, mg/dL	1.08 (0.87–1.28)
GFR, ml/min/1.73 m^2^	60.0 (44.5–70.0)
Hematocrit, %	38.9 (34.4–42.2)
Hemoglobin, g/L	12.9 (11.4–14.3)
WBC, ×10^3^/µL	7.1 (5.7–8.4)
MSCT parameters	
LFA diameter, mm	5.7 (4.8–6.8)
RFA diameter, mm	5.4 (4.9–7.1)
LCCA diameter, mm	6.2 (5.8–7.2)
RCCA diameter, mm	7.1 (5.9–8.1)
BAV	9 (27.2)
LM height, mm	12.7 (11.5–14.7)
RCA height, mm	17.7 (15.2–19.6)
Annulus perimeter, mm	444 (390.3–492.4)
AVAA, mm^2^	76.6 (70.9–79.7)

Data are presented as the median (Q1–Q3) or as *n* (%). TC-TAVI = transcarotid transcathether aortic valve implantation, BMI = body mass index, NYHA = New York Heart Association, COPD = chronic obstructive pulmonary disease, PAD = peripheral arterial disease, DAD = descending aortic disease, MI = myocardial infarction, PCI = percutaneous coronary intervention, TIA = Transient Ischemic Attack, GFR = glomerular filtration rate, MSCT = multi-slices computed tomography LFA = left femoral artery, RFA = right femoral artery, LCCA = left common carotid artery, RCCA = right common carotid artery, BAV = bicuspid aortic valve, LM = left main coronary artery, RCA = right coronary artery, AVAA = aortic valve annulus area.

**Table 2 jcm-10-01499-t002:** Echocardiographic characteristics.

	Pre-Procedural	Post-Procedural	Follow-Up
LVEF, %	50.0 (45–55)	55.0 (50–60)	55.0 (45–60)
Pg max, mmHg	69.0 (59–89)	15 (12.6–18.7) *	15 (12–19) *
Pg mean, mmHg	42.0 (35.5–54.5)	8 (6–10) *	8 (6–9) *
Vmax, m/s	4.1 (3.8–4.6)	1.9 (1.7–2.1) *	1.9 (1.7–2.0) *
AVA, cm^2^	0.7 (0.6–0.8)	1.8 (1.5–1.9) *	1.8 (1.5–2.0) *
PVL ≥ 2 grade	-	2 (6.25)	1 (3.2)

Data are presented as the median (Q1–Q3) or as *n* (%). LVEF = left ventricular ejection fraction, Pg max = aortic valve maximal gradient, Pg mean = aortic valve mean gradient, AVA = aortic valve area, Vmax = transaortic peak instantaneous velocity. * statistically significant differences from baseline values (*p* < 0.001).

**Table 3 jcm-10-01499-t003:** Characteristics of the valve prostheses.

Prosthesis Type	Prosthesis Size (*n*)	Sheath Size (*n*)	Sheatless (*n*)
MedtronicEvolut R	26 (8)	14 (2)	6
MedtronicEvolut R	29 (13)	14 (5)	8
MedtronicEvolut R	34 (3)	16 (1)	2
Portico	27 (1)	16 (0)	0
Edwards-Sapien 3 Ultra	23 (3)	14 (3)	0
Edwards-Sapien 3 Ultra	26 (4)	14 (4)	0

The size of the vascular sheath (French). Edwards-Sapien 3 Ultra (Edwards Lifesciences Corp., Irvine, CA, USA), Evolute R (Medtronic, Minneapolis, MN, USA), Portico (Abbott Vascular, Santa Clara, CA, USA).

**Table 4 jcm-10-01499-t004:** Follow-up.

	TC-TAVI (*n* = 33)
Perioperative	
Procedural success	32 (96.9)
Procedural mortality	1 (3.0)
Procedural time, min.	65 (60–80)
General anaesthesia	33 (100)
30-day outcomes	
Death	2 (6.1)
TIA	1 (3.0)
Stroke	0 (0)
Myocardial infarction	1 (3.0)
Permanent pacemaker implantation	5 (15.1)
Long-term follow-up (19.5 ± 9.5 months)	
Death	7 (21.7)
NYHA class	1(1–2)
NYHA I	22 (71.0)
NYHA II	9 (29.0)
NYHA III	0 (0)
NYHA IV	0 (0)
TIA	2 (6.1)
Stroke	2 (6.1)
Re-hospitalization	4 (12.2)
Carotid Doppler ultrasound	
LCCA PSV, cm/s	71.6 (61.2–86.7)
LCCA EDV, cm/s	9.1 (8.3–14.1)
RCCA PSV, cm/s	70.6 (61.8–83.1)
RCCA EDV, cm/s	10.1 (3.2–16.9)

Data are given as the median (Q1–Q3), mean (SD) or as *n* (%). TC-TAVI = transcarotid transcathether aortic valve implantation, LCCA = left common carotid artery, TIA = transient ischemic attack, ICU = intensive care unit, NYHA: New York Heart Association, GFR = glomerular filtration rate, WBC = white blood cells, PSV = Doppler peak systolic velocity, EDV = Doppler end-diastolic velocity, RCCA = right common carotid artery.

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
