# Peer review of "Long-Term Clinical Outcomes and Carotid Ultrasound Follow-Up of Transcarotid TAVI. Prospective Single-Center Registry"

_jcm, 2021, doi:10.3390/jcm10071499_

Round 1

Reviewer 1 Report

Well executed study and well written manuscript.

As authors mentioned - its single center, small study: cannot draw any definitive conclusions from this study. 

Reviewer 2 Report

The authors investigated the feasibility, safety and long term outcomes of transcarotid TAVI.

The main findings of this study is that cerebral oximetry-guided transcarotid access is safe, has a favorable long-term outcome, and does not increase the risk of plaque formation in the carotid artery.

The topic of this study potentially interesting and clinically relevant and the paper is well written. I have an interest of this study. How did you select the kind of THV? The authors should add this information and discuss about it. It can make this manuscript more interesting .

Reviewer 3 Report

The study carried out by Hudziak et al. shows that TC-TAVI is a safe approach, with favorable long-term outcomes, and with no increased risk of plaque formation in carotid arteries.

The topic is interesting. Anyway, the main limitation of this study is that a control group (TF-TAVI patients) is missing.

Major comments

1) Please, provide an extensive review of English throughout all the text;

2) Could you explain in which cases a transesophageal echo was deemed necessary?

3) Before asserting that TC-TAVI can be regarded as first choice alternative to TF-TAVI (as I can read in the abstract), detailed data on other approaches should be analyzed and reported.

Minor comments

1) The acronym “PAD” is mentioned for the first time in the text on page 2, line 67. Please, specify the meaning; viceversa, on page 4, line 125 and 128 you can leave “PAD”;

2) Replace “transfemoral TAVI” with “TF-TAVI” on page 2, line 69;

3) The acronym “CCA” is mentioned for the first time in the text on page 2, line 73. Therefore, the meaning has to be specified;

4) The same of previous points for the acronym ECG (page 2, line 86);

5) Replace “TC TAVI” with “TC-TAVI” on page 4, line 116, and “TF TAVI” with “TF-TAVI” on page 9, line 241 and 273;

6) Please, write the meaning of the acronym “TIA” on page 6, line 172;

7) Replace [aortic valve area (AVA), aortic valve mean gradient (PG mean), aortic valve maximal gradient (PG max), and transaortic peak instantaneous velocity (Vmax)] with [AVA, mean PG, maximal PG, and transaortic peak instantaneous velocity (Vmax)] on page 6, line 187,188;

8) Please, the acronym “PVL” on page 8, line 218;

9) Delete “peak systolic velocity” and “end diastolic velocity” on page 8, lines 221,222. The acronyms have already been mentioned before in the text;

10) Decide if you want to write “Heart Team” or “heart team” in the text, and change according to your decision.

Reviewer 4 Report

Review for “Long-term clinical outcomes and carotid ultrasound follow-up 2

of transcarotid TAVI. Prospective single-center registry“ by Hudziak et.al.

march 22nd 2021

Summary: The presented study by Hudziak et al. describes a cohort of 33 patients which underwent transcarotid transcatheter aortic valve implantation under cerebral oxymetric control. The patients were followed- up for a mean period of 19.5 months regarding major adverse events with follow-up echocardiographic and clinical data.

The study is well-written and informative, since TC-TAVI is rarely employed yet a valid alternative to transfemoral or transapical approaches.

My comments are as follows:

Abstract:

- Consider explaining TAVI and TIA acronyms, even if they are known.

- Since there seem to be no lost to follow up, state the survival rate as the real percentage of survived patients, 84% without standard error, since there is no need for a estimate if you observed the events. In Figure 3, 2yr survival estimate is stated as 83%.

- line 26: “it can be regarded as a first-choice-alternative to TF access” seems a bit of overstatement to me, as unfortunately TC-TAVI is the least described in literature with the smallest number of patients included in studies

Text:

In my opinion, statement of NYHA classes should be made as categorical, showing how many pts were in class I, II,III, IV with percentages and accordingly, Wilcoxon Tests should be used for follow up. Median NYHA reporting is confounding.This applies for the text and tables.

Please add the annual ratio of TC/TF TAVI implantation rates of the center.

PVL acronym should be written in extenso once.

Periprocedural outcomes:

Line 177 – it is difficult to state a p=0.058 as “no difference”. Change to “no significant difference”, since this is a trend and likely to be significant with bigger sample size.

Limits have been addressed adequately and consent form statement has been made.

Round 2

Reviewer 3 Report

No further editing is required.